# OpenReview forum: "Dissecting Adversarial Robustness of Multimodal LM Agents"
_ICLR.cc/2025/Conference — ICLR 2025 Poster_

### Official Review · Reviewer_igyD · 2024-10-27

**Soundness:** 3
**Presentation:** 3
**Contribution:** 2
**Rating:** 6
**Confidence:** 3

**Summary:**

This paper explores the adversarial robustness of multi-modal LLMs in a realistic threat model built upon VisualWebArena. The created Agent Robustness Evaluation (ARE) framework uses a graph to show the information flow and assesses agent vulnerabilities across text and visual modalities. The paper pays special attention to how the robustness of agents changes concerning individual components. Furthermore, the framework will be accessible for stress testing after the review period.

**Strengths:**

The evaluation framework is novel and addresses a high-impact topic as the importance of multimodal agents continues to grow. The curation of 200 adversarial tasks and evaluation functions is a great contribution to accelerate security research in this field and the paper offers valuable insights into agent vulnerabilities.

**Weaknesses:**

Spelling errors (e.g. We instead of Web in the abstract)

Limited modalities (text, image). An extension to video and sound would be great for future work.

The paper addresses the setting of the VisualWebArena. Further studies in other settings would help for more generalisable insights. The same applies to investigating multi-agent interactions and agents that use function calling.

The authors themselves call the used defences “simple”. Investigation of stronger defences would be of interest to show the extent of vulnerability of an organisation using state of-the-art methods.

**Questions:**

Could the authors elaborate on how ARE might be adapted or extended to other kinds of environments beyond the web-based scenarios tested?

What are the performance losses that are caused by various defences? Which ones are hypothesized by the authors to offer the based protection/performance trade-off?

The performance of a state-of-the-art defence setup on ARE would be of interest

---

> ### Author Response · Authors · 2024-11-18
>
> Thank you for your detailed review! Here are our replies to each of your questions:
>
> > **Review:** Could the authors elaborate on how ARE might be adapted or extended to other kinds of environments beyond the web-based scenarios tested?
>
> Since our ARE framework does not assume the modality or environment, it can be applied to other environments/tasks, e.g., VideoWebArena for the video modality and OSWorld for computer control. ARE can also be naturally extended to the multi-agent setting, where the graph of components can be extended to the graph of specialized agents.
>
> > **Review:** What are the performance losses that are caused by various defenses? Which ones are hypothesized by the authors to offer the based protection/performance trade-off?
>
> We see that the explicit inconsistency check defense and the newly added paraphrase defense achieve the benign performance of 74%, same as the no defense setting, while the two system prompt-based defenses decrease the benign performance to 60%. Overall, the explicit inconsistency check defense has the best protection/performance trade-off (same benign performance, and almost zero ASR). However, as we discussed in the paper, the explicit inconsistency check might not be desirable in practice since it largely increases the number of API calls by checking each image’s text description (e.g., 70% of webpages in our evaluations have more than 10 images). Furthermore, the API-based LM can also be broken by our CLIP attack with an ASR of 38%.
>
> > **Review:** The performance of a state-of-the-art defense setup on ARE would be of interest
>
> Our paper includes a range of defenses and, in response to your comment, we’ve added more. These generally span the "state-of-the-art" defenses proposed in the LLM space that can be applied to large black-box models. We also adapted these defenses to our web agent setting with non-trivial efforts (e.g., rewriting the prompts, deciding which part of input to paraphrase). Here are all the defenses in the updated paper:
>
> -  *Data delimiters + system prompts* [1] instructs the LM to ignore instructions in the data section of the context window. We tried two different prompts (one instructs the LM to ignore the adversarial instructions and one instructs the model to abstain) but found no improvements in security – the attack success rate (ASR) is around 30% before and after the defense.
>
> - *Instruction hierarchy (or StruQ)* [2,3] is a training method that defends the LM from being distracted by untrusted inputs. Although the original paper from OpenAI [1] didn’t release any model, our best guess is that this is probably used for training GPT-4o but not GPT-4V (based on the timing of their publication). We indeed see that GPT-4o is significantly more robust than GPT-4V (67% vs 31% ASR), suggesting instruction hierarchy is helpful in the agent setting. However, the absolute attack success rate on GPT-4o is still high (31% ASR), which means instruction hierarchy has not solved the problem.
>
> - We added the *paraphrase defense* [4], where the untrusted text input to the LM is paraphrased by GPT-4o. The hope is that some adversarial text designed to distract LMs (e.g., “THIS IS VERY IMPORTANT”) will be more benign after paraphrasing. We see that the paraphrasing defense can slightly lower the ASR from 31% to 27.5%. This defense is better than system prompts but still doesn't quite work.
>
> To summarize, defenses without training the LMs show no or limited effectiveness against the attacks. Instruction hierarchy shows promising results by lowering the ASR by half, but still leaves a high ASR of 31%.
>
> [1] https://arxiv.org/abs/2403.14720
>
> [2] https://arxiv.org/abs/2404.13208
>
> [3] https://arxiv.org/abs/2402.06363
>
> [4] https://arxiv.org/abs/2309.00614
>
> Thank you again for these insightful comments! We believe these changes based on your feedback largely strengthen the paper. We hope we have addressed all your concerns and look forward to further feedback.

---

> > ### Comment · Reviewer_igyD · 2024-11-19
> >
> > Thank you for the detailed response. I will increase my confidence rating that your paper is above the acceptance threshold.

---

### Official Review · Reviewer_WdHH · 2024-11-04

**Soundness:** 3
**Presentation:** 2
**Contribution:** 2
**Rating:** 6
**Confidence:** 3

**Summary:**

The paper proposes attack method against multi-modal agent. It considers four types of agent framework and different attack scenarios (white-box. black-box).

**Strengths:**

The paper is well-written and easy to follow. Evaluations are comprehensive.

**Weaknesses:**

The contribution of the paper lies more on the construction of the robustness evaluation framework for agent framework. More discussions on the technical challenges of this would be beneficial to emphasize the contribution.

**Questions:**

Please refer to the weakness part.

---

> ### Author Response · Authors · 2024-11-20
>
> Thank you for your review. We are glad that you recognized our agent robustness framework as a valuable contribution. We also appreciate your concern that the technical challenges were not emphasized sufficiently. To address this, we would like to highlight the challenges from two key perspectives: attack methods and benchmark construction.
>
> **Technical challenges on attack methods**
>
> We put significant efforts into making attacks work for agents based on black-box frontier LMs. We have provided details of the key changes and their ablations in Section 4.3, Section 5.1 and Appendix C. Some of the observations are, to the best of our knowledge, quite novel and useful, e.g., (1) the effectiveness of CLIP model ensembles for targeted adversarial attacks (Section 5.1, Appendix C.1), (2) lower optimization resolution improves the CLIP attack (Appendix C.1), (3) insights on when adversarial attacks on a single image generalize to the image embedded in a larger visual context, and how to improve it (Section 5.1, Appendix C.2).
>
> **Technical challenges on benchmark construction**
>
> A significant part of our contribution is the VWA-Adv benchmark. In our benchmark, evaluation functions require careful and manual design. This is because we used a real environment (web environment that runs with Chrome), where tasks and evaluation functions cannot be procedurally generated. We’ve now added a more detailed description of them in Appendix A.2. Specifically, we manually construct an evaluation function for each task, using the evaluation primitives from Koh et al.2024 . Once the agent terminates, the evaluation function evaluates the environment state and/or the agent’s response to determine whether the adversarial goal has been achieved. Examples of evaluation functions include: (1) checking if a target product is added to the cart, (2) verifying if the agent has navigated to the target webpage, (3) checking if the agent has submitted target text (fuzzy-matched by GPT-4) in a target form, and (4) checking whether the agent’s response contains specific target text.
>
> Thank you again for your comments! We hope these changes/replies address your concerns. If there are specific aspects of our novelty/significance that we can address to help your evaluation of our paper, we’d be happy to discuss them further.

---

> > ### Author Response · Authors · 2024-11-24
> > **Follow-up on review**
> >
> > As the discussion deadline approaches, we wanted to follow up and see if there are any additional questions or concerns that we might address.

---

> > ### Comment · Reviewer_WdHH · 2024-11-29
> >
> > Thanks for the clarification! I highly suggest the authors highlight these efforts in revisions and adjust my score accordingly.

---

> > > ### Author Response · Authors · 2024-11-30
> > >
> > > Thank you for your suggestion! Since we’ve passed the deadline (Nov 27) for uploading a new pdf, we are happy to highlight them in the final paper, especially those in the Appendix currently.

---

### Official Review · Reviewer_G5oP · 2024-11-04

**Soundness:** 2
**Presentation:** 2
**Contribution:** 2
**Rating:** 5
**Confidence:** 4

**Summary:**

This paper examines the vulnerability of multimodal language model agents to adversarial attacks, particularly in the context of web-based environments. The authors create a framework, Agent Robustness Evaluation (ARE), to systematically analyse the robustness of different agent components and their interactions. They find that even state-of-the-art agents, including those using GPT-4 and incorporating advanced techniques like reflection and tree search, are susceptible to attacks.

**Strengths:**

The framework models agents as graphs, with each node representing an agent component and each edge representing the flow of information between components. ARE decomposes the final attack success into edge weights that measure the adversarial influence of information propagated on the edge. This framework could allow researchers to understand the robustness/vulnerability of various components and agent configurations.

The paper shows some baseline defenses based on prompting and consistency checks and finds that they offer limited gains against attacks. This finding highlights the need for more research on robust defenses for multimodal agents.

**Weaknesses:**

The main weakness of the submitted paper is that it does not clearly motivate and explain why the proposed framework is representative of a real scenario and the usage of integrated LLMs. Although we can, of course, not expect to model the real world perfectly, it remains unclear why the proposed framework is a good approximation.

**The study focused on a specific threat model** where the attacker is a legitimate user of the platform with limited capabilities. This might not encompass the full range of potential threats that agents could face in real-world deployments. For example, attackers with more sophisticated capabilities or access to internal systems might be able to bypass the limitations considered in this study and launch more effective attacks.

**The defenses explored in the paper are quite basic** and show limited effectiveness against the attacks. While they offer some insights into potential defense strategies, more sophisticated and robust defenses are needed to mitigate the vulnerabilities identified in the paper effectively.

**The attacks presented in the paper rely on well-engineered versions of existing attacks**, adapted to the agent setting. This means that the true risk of these attacks could be higher as more advanced and novel attack strategies are developed.

**Questions:**

See above

---

> ### Author Response · Authors · 2024-11-19
>
> Thank you for your review. We appreciate the concerns raised, most of which also align with the challenges we openly addressed in the Limitations section (Appendix D). Most of them are deliberate design choices we made to focus on (1) realistic threat to agent robustness and (2) understanding the contribution of components to the robustness of agent systems. Here are our replies to each of your questions:
>
> > **Review:** The study focused on a specific threat model where the attacker is a legitimate user of the platform with limited capabilities.
>
> We would like to emphasize that the threat model we consider is both common and realistic, based on our understanding of real-world scenarios. The assumption of limited environment access reflects practical constraints often observed in real-world applications, where the service providers can devote **significant resources** to **limited direct access** to hijack the platform. It also makes our observation more concerning – despite the attacker’s limited access, the attack success rate is already high.
>
> > **Review:** The defenses explored in the paper are quite basic
>
> Our paper includes a range of defenses and, in response to your comment, we’ve added more. These generally span the "state-of-the-art" defenses proposed in the LLM space that can be applied to large black-box models. We also adapted these defenses to our web agent setting with non-trivial efforts (e.g., rewriting the prompts, deciding what input to paraphrase). Here are all the defenses in the updated paper:
>
> -  *Data delimiters + system prompts* [1] instructs the LM to ignore instructions in the data section of the context window. We tried two different prompts (one instructs the LM to ignore the adversarial instructions and one instructs the model to abstain) but found no improvements in security – the attack success rate (ASR) is around 30% before and after the defense.
>
> - *Instruction hierarchy (or StruQ)* [2,3] is a training method that defends the LM from being distracted by untrusted inputs. Although the original paper from OpenAI [1] didn’t release any model, our best guess is that this is probably used for training GPT-4o but not GPT-4V (based on the timing of their publication). We indeed see that GPT-4o is significantly more robust than GPT-4V (67% vs 31% ASR), suggesting instruction hierarchy is helpful in the agent setting. However, the absolute attack success rate on GPT-4o is still high (31% ASR), which means instruction hierarchy has not solved the problem.
>
> - We added the *paraphrase defense* [4], where the untrusted text input to the LM is paraphrased by GPT-4o. The hope is that some adversarial text designed to distract LMs (e.g., “THIS IS VERY IMPORTANT”) will be more benign after paraphrasing. We see that the paraphrasing defense can slightly lower the ASR from 31% to 27.5%. This defense is better than system prompts but still doesn't quite work.
>
> To summarize, defenses without training the LMs show no or limited effectiveness against the attacks. Instruction hierarchy shows promising results by lowering the ASR by half, but still leaves a high ASR of 31%.
>
> > **Review:** The attacks presented in the paper rely on well-engineered versions of existing attacks, adapted to the agent setting. This means that the true risk of these attacks could be higher as more advanced and novel attack strategies are developed.
>
> We'd like to mention that this is a point we had openly addressed in the limitations section (Appendix D): ***First, our attack baselines are well engineered versions of existing attacks, with necessary modifications to the agent setting. While some of them show strong attack success, they only serve as the lower bound of risks.*** We wrote this with the intention that the true risks could be higher as new attacks are developed in the future, and our benchmark can help keep track of the progress of both attacks and defenses.
>
> However, we want to emphasize that we have put significant efforts into making existing attacks work for agents based on black-box frontier LMs. We have provided details of the key changes and their ablations in Section 4.3, Section 5.1 and Appendix C. Some of the observations are, to the best of our knowledge, quite novel and useful, e.g., (1) the effectiveness of CLIP model ensembles for targeted adversarial attacks (Section 5.1, Appendix C.1), (2) lower optimization resolution improves the CLIP attack (Appendix C.1), (3) insights on when adversarial attacks on a single image generalize to the image embedded in a larger visual context, and how to improve it (Section 5.1, Appendix C.2).
>
> [1] https://arxiv.org/abs/2403.14720
>
> [2] https://arxiv.org/abs/2404.13208
>
> [3] https://arxiv.org/abs/2402.06363
>
> [4] https://arxiv.org/abs/2309.00614
>
> Thank you again for your review! We believe these changes and clarifications based on your feedback largely strengthen the paper. We hope we have addressed all your concerns and look forward to further feedback.

---

> > ### Comment · Reviewer_G5oP · 2024-11-21
> >
> > I would be happy to increase my score based on the additional effort you made. At the moment, however, the threat scenario remains unclear. I had another look into the described threat scenario. Mainly based on Figure 1 and the description provided in Section 3.1: From a computer system point of view, how should it be possible to write a comment, if the goal is to add a product to a chart? Maybe these questions can answer to make clear what is unclear:
> >
> > - What system are we describing here? Is it an already existing tool that allows us to control an online shop? Or is is an hypothetical tool. In the first case, please make sure to which tool you refer. In the second case, add an explanation of the attacked system
> > - What is the knowledge of the attack? Do they know how the trigger image will be used and how? Please explain this in detail how the image is placed and by whom
> > - What is the attacker capable of? this is very related to the point above, but explains what the attacker has access to to get to their goal.

---

> > > ### Author Response · Authors · 2024-11-22
> > >
> > > Thank you for looking at our response and your further feedback! We are glad that the efforts we made helped your evaluation on our paper. At the high level, we curated our adversarial tasks and evaluation functions based on the VisualWebArena environment (Koh et. al. 2024). VisualWebArena has three individual, web-based environments: classifieds, online forums, and shopping websites. Each environment is a sandbox built upon an open-source web template. We’d like to use **the classifieds environment as an example**, but the same logic applies to the other two.
> > >
> > > For the classifieds environment, we used OsClass classified script as a sandbox: https://osclass-classifieds.com/ In our experiments, the environment is opened as a website in Chrome, and agents can interact with it using the same action space as humans (e.g., click, type, scroll). An official (not affiliated with us) demo of this website is at https://demo.osclasspoint.com/en/ Here are our responses to your specific questions:
> > >
> > > > In Figure 1, from a computer system point of view, how should it be possible to write a comment, if the goal is to add a product to a cart?
> > >
> > > The environment provides different interactive options, e.g., add to cart, contact seller, leave comments, etc. The threat model in Figure 1 shows that the attacker can redirect the agent to pursue a different goal that requires using other available options. Here is an example of a page with the commenting option: https://demo.osclasspoint.com/en/for-sale/electronics/acer-predator-geforce_i17
> > >
> > > > What system are we describing here? Is it an already existing tool that allows us to control an online shop? Or is it a hypothetical tool. In the first case, please make sure to which tool you refer. In the second case, add an explanation of the attacked system
> > >
> > > In our classifieds environment, for example, we focus on a malicious seller attacking a buyer’s AI agent on a classifieds website built by the OsClass classified script (https://osclass-classifieds.com/). The agent browses the website just like humans (input is the current screen, and output is the next action to take). The attacker has access to their own product images/descriptions and wants to manipulate the agent behavior to be in favor of their own product.
> > >
> > > > What is the knowledge of the attack? Do they know how the trigger image will be used and how? Please explain this in detail how the image is placed and by whom
> > >
> > > In our classifieds environment, for example, the seller (i.e., attacker) has access to their own product images/descriptions (i.e., trigger images/texts). They know that their trigger images/texts will be part of the agent’s input space; however, they **do not know** what exactly other products are around their product or what prompt the AI agent is using.
> > >
> > > > What is the attacker capable of? this is very related to the point above, but explains what the attacker has access to to get to their goal.
> > >
> > > The seller (i.e., attacker) has access to their own product images/descriptions (i.e., trigger images/texts), and they can only change them – they cannot make arbitrary changes to others’ products or the website layout as they typically do not have the access.
> > >
> > > Thank you again for your follow-up questions! We hope we have addressed your concerns and please let us know if anything remains unclear.

---

> > > > ### Author Response · Authors · 2024-11-24
> > > > **Follow-up on review**
> > > >
> > > > As the discussion deadline approaches, we wanted to follow up and see if there are any additional questions or concerns that we might address.

---

> > > > > ### Comment · Reviewer_G5oP · 2024-11-26
> > > > >
> > > > > Thank you again for the clarification. I have adjusted my score. Unfortunately, the full contributions are not enough. I would encourage you to reflect a little better in the paper on the actual setting we are operating in to make the contribution easier to understand.

---

> > > > > > ### Author Response · Authors · 2024-11-28
> > > > > >
> > > > > > Thank you for your suggestion. In the updated paper, we provided (1) a more detailed description of our setting in Section 3.1 with examples, and (2) visual illustration of the agent's observation space in the appendix.

---

### Official Review · Reviewer_BhGk · 2024-11-04

**Soundness:** 3
**Presentation:** 2
**Contribution:** 2
**Rating:** 8
**Confidence:** 4

**Summary:**

This papers studies the robustness of multimodal language model agent **systems** to attacks which attempt to force the agent to take arbitrary actions. The authors introduce the concept of agent graph, which is a graph whose nodes are the system components and the edges represent the flow of data from one system component to the other, and the weight of each edge represents how likely is an attack to succeed after the given component. This graph allows a more comprehensive understanding of how vulnerable each component is and whether it robustifies or weakens the system. Using this approach, the authors studies the robustness of the different components uner several attacks both text-only and image-based against several models, and also show the effectiveness (or lack thereof) of different defenses.

**Strengths:**

- The authors look at the entire deployed system and not only at the LLM component of the system. This is very important as the individual components of the system might amplify or reduce robustness (as shown by the paper).
- While at the beginning feels a bit of an unnecessary formalization, I ended up liking describing the system as a directed graph where the weights of the nodes represent the likelyhood of success of the attack after the given component.
- VWA-Adv seems to be a good dataset to evaluate the robustness of agents in an adversarial environment and the design choices are sound.
- The threat modeling is realistic.

**Weaknesses:**

- The main complain I have is that there is no discussion whatsoever regarding (indirect) prompt injection attacks [1] and all the related literature. Prompt injection attacks are very relevant as they use a similar attack vector (an adversary manipulating untrusted data) which has the same aim (trigger some specific actions). The paper would benefit from a discussion of prompt injection attacks and an explanation on how "Text access" attacks differ from those.
- I find the caption of the figures too short. I believe they could benefit from more detailed descriptions of the figures. For example, the caption for Figure 3 could include an explanation of what happens in each of the subfigures. What does the arrow with "1.0" in the bottom right part of Figure 3.c mean? Is it an attack?
- How are tasks evaluated? The paper mentions "We employ evaluation primitives from Koh et al. (2024a) and manually annotate the evaluation function.", but I believe that these details should be included in this paper too (at least in the appendix) to make it self-contained.
- In general, the narrative of "x improves security if kept uncompromised, but it decreases it if compromised" slightly dangerous by giving a false sense of security. When evaluating the security of a system, one should consider the worst case scenario. So, while I find these insights valuable, as they can be useful to understand where to focus when coming up with a defense, I believe that they should rephrased differently to reflect that the fact that, in fact, they are insecure.

References

[1] https://arxiv.org/abs/2302.12173

**Questions:**

See weaknesses.

---

> ### Author Response · Authors · 2024-11-18
>
> Thank you for your detailed review! We are happy to see that you generally like our framework and benchmark. We also appreciate your constructive feedback, and here are our replies to each of them:
>
> > **Review:** No discussion regarding (indirect) prompt injection attacks. The paper would benefit from a discussion of prompt injection attacks and an explanation on how "Text access" attacks differ from those.
>
> We acknowledge that the definition of "text injection attack" is indeed equivalent to "prompt injection attack", and we did not intend to claim novelty here. Our intention was to study the difference in the two modes of text vs image. In the updated paper, we have replaced “text injection attack” with "prompt injection attack" to solve the ambiguity of existing work; however, we still keep “text vs image access” to make the two settings clear.
>
> To further clarify how our prompt injection work fits into the broader literature: prompt injection is a general framework where either malicious instructions or misleading information are used to compromise LLMs/agents. Different attacks in prior work tend to differ in terms of how the attackers access the LLMs, and what harmful behavior they target. In our work, we focus on the web-agents setting in a realistic threat model, and also focus on how the prompt injection attack’s success varies depending on the configuration surrounding the agent. We also study the difference between attacking the text space directly vs via the images (this would constitute an indirect prompt injection attack in the terminology of the paper you referenced). We have edited and bolstered our related work section to better contextualize our prompt injection attack and work in the broader literature on prompt injection. We hope this makes things clear, and let us know if there are further suggestions/questions.
>
> > **Review:** The caption of the figures too short. For example, the caption for Figure 3 could include an explanation of what happens in each of the subfigures. What does the arrow with "1.0" in the bottom right part of Figure 3.c mean? Is it an attack?
>
> We have expanded the captions throughout the paper – thank you for the suggestion! Updated caption for Figure 3: *Adding a new component to an agent can either improve or harm robustness. If $B$ only receives input (if any) from the trusted environment, $B$ would lower $\lambda$. However, an attacker can also attack this new component (introducing an edge of weight $1$) that could increase $\lambda$.* To answer your question, the arrow with "1.0" means component B takes adversarial inputs from a source other than A, which could be a direct attack from the environment or intermediate outputs from another component.
>
> > **Review:** How are tasks evaluated?
>
> The evaluation functions are indeed important in the benchmark, and we’ve now added a more detailed description of them in Appendix A.2. Specifically, we manually construct an evaluation function for each task, using the evaluation primitives from Koh et al.2024 . Once the agent terminates, the evaluation function evaluates the environment state and/or the agent’s response to determine whether the adversarial goal has been achieved. Examples of evaluation functions include: (1) checking if a target product is added to the cart, (2) verifying if the agent has navigated to the target webpage, (3) checking if the agent has submitted target text (fuzzy-matched by GPT-4) in a target form, and (4) checking whether the agent’s response contains specific target text.
>
> > **Review:** In general, the narrative of "x improves security if kept uncompromised, but it decreases it if compromised" slightly dangerous by giving a false sense of security. When evaluating the security of a system, one should consider the worst case scenario.
>
> We agree with your "worst-case scenario" perspective and have addressed this in the updated paper. Our intent was to highlight exactly what you brought up, and we apologize for the poor phrasing. For example, the updated description on the evaluator is: *Reflexion agents with an uncompromised/robust evaluator can self-correct attacks on policy models. However, this creates a false sense of security – in the worst-case scenario, an evaluator can get compromised and decrease robustness by biasing the agent toward adversarial actions through adversarial verification and reflection.* We think the nuanced distinction between “compromised” vs “uncompromised” could be beneficial for informing the design of agents—we should pay special attention to evaluators and the evaluator should be added to a system only if it can be made robust / kept compromised. We hope this new phrasing addresses your concerns and better describes the takeaways from our work.
>
> Thank you again for these insightful comments! We believe these changes based on your feedback largely strengthen the paper. We hope we have addressed all your concerns and look forward to further feedback.

---

> > ### Comment · Reviewer_BhGk · 2024-11-19
> >
> > Thanks for addressing my concerns. I am largely satisfied with the changes you made in the revision. Just one further comment on the Reflexion agents: the abstract (in particular the sentence "For example, an evaluator and value function, if kept uncompromised/robust, can decrease the attack success rate (ASR) by 22% and 17%, relatively, but if compromised, can increase the ASR by 15% and 20%, relatively." still reads in a sub-optimal way imo, and it would benefit from a rephrasing too. I will update my score, and I will be happy to further update if this concern will be addressed (I believe it's quite important to convey the right messages in the abstract as many people will just stop there and draw conclusions from it!)

---

> ### Author Response · Authors · 2024-11-19
>
> Thank you for looking at our response and for your further feedback! We are glad that the changes we made addressed most of your concerns. Regarding the abstract, here is an updated version:
>
> > We find that *new components that typically improve benign performance can open up new vulnerabilities and harm robustness*. An attacker can compromise the evaluator used by the reflexion agent and the value function of the tree search agent, which increases the attack success relatively by 15% and 20%.
>
> In the updated abstract, we try to convey a more straightforward message about the scenario where the evaluator/value function is compromised, and leave the more nuanced results to the intro/experiment sections. If there are any further suggestions you think would improve it, we’d be happy to iterate over them!

---

> > ### Comment · Reviewer_BhGk · 2024-11-20
> >
> > Thank you for the update, I will further increase my score as I believe that this paper should be accepted

---

### Meta-Review · Area_Chair_eDRd · 2024-12-13

**Metareview:**

The recommendation is based on the reviewers' comments, the area chair's evaluation, and the author-reviewer discussion.

This paper proposes the Agent Robustness Evaluation (ARE) framework. All reviewers find the studied setting novel and the results provide new insights. The authors’ rebuttal has successfully addressed the major concerns of reviewers.

In the post-rebuttal discussion, most reviewers suggested acceptance, while one review was still concerned about the clarity of the considered threat model. I deem this comment addressable and ask the authors to make mandatory changes to improve clarity.

Overall, I recommend acceptance of this submission. I also expect the authors to include the new results and suggested changes during the rebuttal phase to the final version.

**Additional Comments On Reviewer Discussion:**

In the post-rebuttal discussion, most reviewers suggested acceptance, while one review was still concerned about the clarity of the considered threat model. I deem this comment addressable and ask the authors to make mandatory changes to improve clarity.

---

### Decision · Program_Chairs · 2025-01-22

Accept (Poster)